# The Burden of Cardiovascular Disease and Geriatric Syndromes in Older Patients Undergoing Chronic Hemodialysis

**DOI:** 10.3390/ijerph21060812

**Published:** 2024-06-20

**Authors:** Tan Van Nguyen, Thu Thi Xuan Pham, Tu Ngoc Nguyen

**Affiliations:** 1Department of Geriatrics & Gerontology, University of Medicine and Pharmacy at Ho Chi Minh City, Ho Chi Minh City 700000, Vietnam; 2Department of Interventional Cardiology, Thong Nhat Hospital, Ho Chi Minh City 700000, Vietnam; 3The George Institute for Global Health, University of New South Wales, Sydney, NSW 2000, Australia; 4Sydney School of Public Health, Faculty of Medicine and Health, The University of Sydney, Sydney, NSW 2006, Australia

**Keywords:** cardiovascular disease, geriatric syndromes, cardio-geriatrics, chronic kidney disease, dialysis, multimorbidity

## Abstract

Background. There is limited evidence on the complexity of cardiovascular disease (CVD) and geriatric syndromes in older patients with end-stage renal disease. Our aims were to (1) examine the prevalence of CVD in older patients on chronic hemodialysis, (2) compare the burden of geriatric syndromes in patients with and without CVD, and (3) examine the impact of CVD on hospitalization. Methods. This prospective, observational, multi-center study was conducted at two dialysis units of two major hospitals in Vietnam. Consecutive older adults receiving chronic hemodialysis were recruited from November 2020 to June 2021. CVD was defined as having one of these conditions: heart failure, ischemic heart disease, or stroke. Participants were assessed for geriatric conditions including frailty, malnutrition, impairment in instrumental activities/activities of daily living, depression, falls, and polypharmacy. Multivariable logistic regression analysis was applied to examine the impact of CVD on 6-month hospitalization, adjusting for age, sex, duration of dialysis, Charlson Comorbidity Index, and geriatric conditions. Results were presented as odds ratios (ORs) and 95% confidence intervals (CIs). Results. There were 175 participants (mean age 72.4 ± 8.5 and 58.9% female). CVD was present in 80% of the participants (ischemic heart disease: 49.7%, heart failure: 60.0%, and stroke: 25.7%). Participants with CVD had a higher burden of geriatric syndromes compared to those without CVD. During the 6-month follow-up, 48.6% of the participants were hospitalized (56.4% of those with CVD vs. 17.1% of those without CVD), *p* < 0.001). CVD independently increased the risk of hospitalization (adjusted OR 3.32, 95% CI 1.12–9.80). Conclusions. In this study, there was a very high prevalence of CVD in older patients undergoing chronic dialysis. Participants with CVD had a higher burden of geriatric syndromes and their risk of 6-month hospitalization increased by three times. There is a need for a multidisciplinary and patient-centered approach to treatment planning for these patients.

## 1. Background

Patients with chronic kidney disease, especially end-stage renal disease, exhibit a very high cardiovascular risk [1,2]. Cardiovascular disease (CVD) is the leading cause of mortality in this population [1,2]. End-stage renal disease significantly impacts cardiovascular health, representing a major source of morbidity and mortality among patients with this condition [1,2]. The interaction between end-stage renal disease and cardiovascular health is complex, with multiple factors contributing to an increased risk of CVD. In patients with end-stage renal disease and receiving hemodialysis, there are many cardiovascular disorders such as left ventricular hypertrophy, endothelial dysfunction, volume overload, and upregulation of the renin–angiotensin–aldosterone system [1,3,4]. The uremic toxins, among other factors such as chronic inflammation and high oxidative stress, contributed significantly to the pathophysiology and accelerated progression of CVD in patients with end-stage renal disease [5,6]. In patients with end-stage renal disease, coronary heart disease is highly prevalent due to multiple cardiovascular risks and comorbidities such as obesity, smoking, reduced physical activity, dyslipidemia, hypertension, and diabetes mellitus [1,2]. The risk of coronary heart disease in patients with end-stage renal disease and initiating dialysis is 5- to 20-fold higher than that of the general population [1,5]. The risk of congestive heart failure is also 12 to 36 times higher compared to the general population, with approximately 36% of all patients with end-stage renal disease having heart failure at the initiation of dialysis [5,7]. Besides common risk factors for heart failure such as advancing age, diabetes mellitus, hypertension, and coronary heart disease, chronic volume overload, poorly controlled hypertension, and uremic toxins may have a negative impact on myocardial contractility and function [1,7]. Chronic kidney disease is also a risk factor for embolic stroke [8,9,10]. Therefore, it is essential to understand the burden of CVD and to provide appropriate management of CVD for this population.

In Vietnam, chronic kidney disease and CVD are prevalent in older people. The CKD incidence was approximately 120 per million people in Vietnam [11]. The burden of chronic kidney disease remains high, with more than 10,000 cases per 100,000 population and a mortality rate of 20.6 per 100,000 [12]. Furthermore, the population afflicted with end-stage renal disease is on an upward trajectory, with an annual increment reaching approximately 90,000 patients [13]. These figures underscore the ongoing and intensifying challenge posed by chronic kidney disease on the healthcare system. However, there are limited studies on the complexity of CVD in older patients with end-stage renal disease and chronic dialysis in Vietnam.

Therefore, in this study in older patients with end-stage renal disease and receiving chronic hemodialysis in Vietnam, we aimed to (1) examine the prevalence of the major CVD, (2) compare the burden of common geriatric syndromes in patients with and without CVD, and (3) examine the impact of CVD on hospitalization.

## 2. Methods

### 2.1. Study Design and Population

This prospective, observational, multi-center study was conducted at two dialysis units of two major hospitals in Ho Chi Minh City, Vietnam. Consecutive older adults (aged 60 or older) diagnosed with end-stage renal disease and on chronic hemodialysis at these two dialysis centers were recruited from November 2020 to June 2021. The exclusion criteria included (1) dementia, (2) having mental illness or visual impairment that could affect their ability to answer the study questionnaires, (3) having acute illnesses that required admission, and (4) not providing consent.

The study was approved by the Ethics Committees of the University of Medicine and Pharmacy in Ho Chi Minh City (Reference Number 788/2020/HDDD-DHYD, 2 November 2020). Informed consent was obtained from all participants. Details of study design and data collection were described in a previous publication [14].

### 2.2. Variable Definition

CVD was defined based on the medical records and as having one of these conditions: heart failure, ischemic heart disease, or stroke.

Comorbidities were assessed using the Charlson Comorbidity Index [15]. The diagnosis of CVD and other comorbidities conformed to the International Classification of Diseases, Tenth Revision.

Geriatric syndromes: participants were assessed for geriatric conditions such as frailty, malnutrition, impairment in activities of daily living (ADL), impairment in instrumental activities of daily living (IADL), depression, falls, and polypharmacy. Frailty was defined according to the Clinical Frailty Scale (CFS), and participants with a CFS total score ≥4 were classified as being frail [16]. Malnutrition was defined as a total score ≤7 from the Mini Nutritional Assessment Short Form [17]. Impairment in activities of daily living was defined as an ADL total score <6 [18]. Impairment in instrumental activities of daily living was defined as an IADL total score <8 [18,19]. History of falls was documented from the medical records. Polypharmacy was defined as the concurrent use of ≥5 medications. Depression was defined based on the 15-item Geriatric Depression Scale (GDS) and participants with a GDS total score ≥10 were classified as having depression [20].

Demographics and lifestyle factors: age and sex were defined as recorded in medical records. Body mass index (BMI, kg/m^2^) was calculated based on measured weight (kg) and height (m). Participants were classified into four groups: underweight (BMI < 18.5 kg/m^2^), normal (BMI 18.5–22.9 kg/m^2^), overweight (BMI 23.0–24.9 kg/m^2^), and obese (BMI ≥ 25.0 kg/m^2^) [21]. Education status was obtained through interviews and included the following categories: illiterate, primary school, secondary school, high school, and higher education (college/university). Smoking status was categorized based on self-report as non-smoking or smoking (including current smokers or ex-smokers who stopped smoking less than one year ago).

All-cause hospitalization: all participants were followed for six months after being included in the study. Information on hospitalization was obtained through medical records and by calling the contact phone numbers provided by participants or their caregivers after six months.

### 2.3. Statistical Analysis

Analysis of the data was performed using SPSS for Windows (version 29.0, IBM Corp., Armonk, NY, USA) and R 4.3.2. Continuous variables are presented as means ± standard deviation, and categorical variables as frequencies and percentages. Comparisons between groups (with and without CVD) were conducted using the Chi-square test or Fisher’s exact test for categorical variables and Student’s *t*-test or Mann–Whitney test for continuous variables.

Multivariable logistic regression analysis was applied to examine the impact of CVD on 6-month hospitalization, adjusted for variables that may have a significant association with hospitalization through clinical judgement, including age, sex, duration of dialysis, Charlson Comorbidity Index, and the geriatric syndromes (frailty, ADL impairment, IADL impairment, falls, malnutrition, depression, and polypharmacy). All variables were checked for interactions. Results were presented as odds ratios (ORs) and 95% confidence intervals (CIs).

## 3. Results

There were 175 participants. They had a mean age of 72.4 ± 8.5, 58.9% were female, 22.9% were underweight, 10.3% were overweight, and 18.9% were obese. CVD was present in 80% of the participants (heart failure: 60.0%, coronary heart disease: 49.7%, and stroke: 25.7%). The mean duration of hemodialysis was 3.6 ± 3.5 years. The mean number of medications used was 7.1 ± 1.9. (Table 1).

Table 2 presents the prevalence of geriatric syndromes in the participants. The most common geriatric syndrome was frailty (87.4%), followed by polypharmacy (81.1%), IADL impairment (81.1%), malnutrition (42.9%), ADL impairment (37.1%), falls (29.1%), and depression (24.0%). Compared to participants without CVD, those with CVD had a significantly higher prevalence of frailty (91.4% vs. 71.4%, *p* = 0.001), IADL impairment (86.4% vs. 60.0%, *p* < 0.001), malnutrition (48.6% vs. 20.0%), *p* = 0.004), and ADL impairment (42.1% vs. 17.1%, *p* = 0.006).

During 6-month follow-up, 48.6% of the participants had at least one hospitalization (56.4% in participants with CVD compared to 17.1% in those without CVD, *p* < 0.001). Figure 1 illustrates the hospitalization rates by CVD and geriatric syndromes. Participants with CVD had a significantly higher all-cause hospitalization rate in 6 months (56.4% vs. 17.1% in participants without CVD, *p* < 0.001). The all-cause hospitalization rate was also higher in participants with malnutrition (66.7% vs. 35.0%, *p* < 0.001), IADL impairment (52.8% vs. 30.3%, *p* = 0.020), and ADL impairment (58.5% vs. 42.7%, *p* = 0.044) compared to those without these conditions.

In the multivariable logistic regression model, CVD independently increased the risk of hospitalization (adjusted OR 3.32, 95% CI 1.12–9.80), allowing for age, sex, duration of dialysis, Charlson Comorbidity Index, and the geriatric syndromes including frailty, ADL impairment, IADL impairment, falls, malnutrition, depression, and polypharmacy. The other factor that was independently associated with increased hospitalization in the adjusted model was malnutrition (adjusted OR 4.30, 95% CI 1.80–10.28) (Figure 2).

## 4. Discussion

This study in 175 older patients undergoing chronic hemodialysis found that the prevalence of CVD was very high (80.0%). Participants with CVD had a higher burden of geriatric syndromes and their risk of 6-month hospitalization increased by approximately three times compared to those without CVD.

Our study findings are in line with the literature, which showed that cardiovascular disease is very common in patients with end-stage renal disease and significantly increased adverse outcomes, including hospitalizations, in this population [1,22]. The management of CVD in older patients with end-stage renal disease can be challenging [1,22]. The unique physiological changes and risk factors associated with end-stage renal disease in combination with aging can cause altered responses to cardiovascular pharmacotherapies [1,3,23]. Cardiovascular medications such as angiotensin converting enzyme inhibitors (ACEIs)/angiotensin receptor blockers (ARBs) and statins may have diminished clinical benefit in patients undergoing chronic hemodialysis [3,23]. In a systematic review and meta-analysis of randomized controlled trials on the effects of ACEIs or ARBs on fatal and nonfatal cardiovascular outcomes in adults undergoing hemodialysis, the authors found that the use of ACEIs or ARBs was not associated with a statistically significant reduction in the risk of cardiovascular events (OR 0.66, 95% CI 0.35–1.25 compared with control, *p* = 0.20) [3]. In a randomized, double-blind, prospective trial including 2776 patients who were on chronic hemodialysis (aged 50 to 80 years), the treatment with rosuvastatin had no significant impact on the composite outcomes of death from cardiovascular causes, nonfatal myocardial infarction, or nonfatal stroke [23]. There should be strategies to increase the participation of patients with end-stage renal disease in randomized clinical trials to examine the optimal cardiovascular management in this population.

Non-adherence to medications is also a common issue in people on chronic hemodialysis [24]. In our study, the mean number of prescribed medications was seven, which was consistent with other reports in patients receiving chronic dialysis [25,26]. In a systematic review on adherence to prescribed oral medication in patients undergoing chronic hemodialysis, the mean non-adherence rate was 67%, with more than half of the 19 included studies reporting 50% or higher non-adherence rates [27].

Our study showed that older patients on chronic hemodialysis had a double burden of CVD and multiple geriatric syndromes. Therefore, multidisciplinary care is essential in managing these patients. There should be a collaboration between nephrologists, cardiologists, geriatricians, dietitians, pharmacists, and other healthcare professionals to develop individualized care plans that address renal, cardiovascular, and aged care needs.

### Strengths and Limitations

To the best of our understanding, this is the first study in Vietnam to investigate the burden of CVD and geriatric syndromes in older patients with end-stage renal disease and undergoing hemodialysis. The study was conducted at two large metropolitan dialysis centers in Vietnam and comprised high-quality clinical data. However, our study had some limitations. The follow-up duration was only six months, and information on the reasons for hospitalization was not well documented. The unavailability of data regarding COVID-19 admissions and tests may have significant implications for the reported admission rates. Additionally, we did not include details on the vaccinations administered, a factor that might contribute to cardiovascular events but remains unexplored within our dataset. Moreover, the study lacks data on alcohol consumption, which is a crucial variable given its potential influence on CVD and geriatric syndrome outcomes. Data on blood pressure measurements were also not collected, depriving us of critical insights into a key risk factor for cardiovascular and other health complications. In the context of dialysis, several important aspects were not recorded. These include the number of dialysis sessions, the vintage of dialysis, and the type of vascular access used. These factors are instrumental in determining dialysis efficacy and the incidence of related complications. Finally, the underlying causes of chronic kidney disease in our study population were not specified. This omission is particularly significant, as different etiologies of chronic kidney disease can have varying impacts on CVD prevalence and the manifestation of geriatric syndromes. By acknowledging these limitations, we can identify areas for future research to enhance the robustness and applicability of our results. Further studies with a longer follow-up duration are needed to understand the common causes for hospitalization in older patients with end-stage renal disease and who are on dialysis. Further studies are also needed to examine medication adherence in older patients receiving chronic dialysis in Vietnam.

## 5. Conclusions

In conclusion, this study demonstrated the high burden of CVD and geriatric syndromes in older patients undergoing chronic hemodialysis and the significant impact of CVD on hospitalization. As the management of CVD in older patients on chronic hemodialysis is complex, there is a need for a multidisciplinary and patient-centered approach to treatment planning. Further research is needed to improve health outcomes for these patients.

## Figures and Tables

**Figure 1 ijerph-21-00812-f001:**
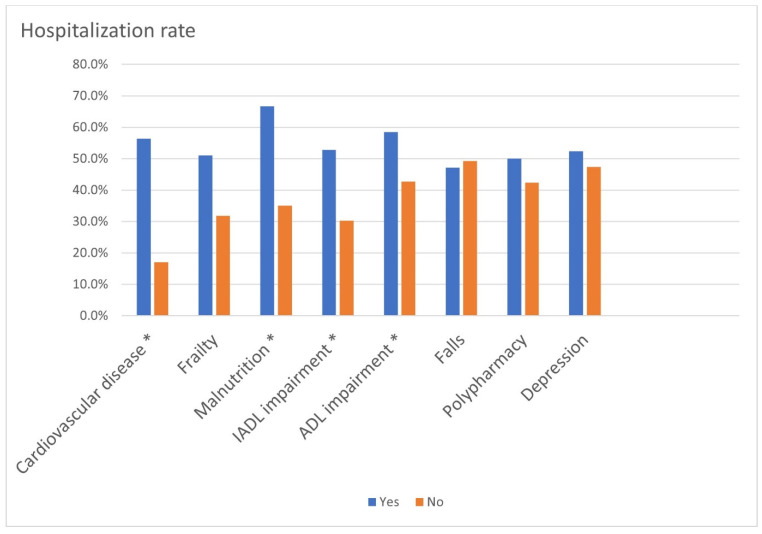
Hospitalization rates by cardiovascular disease and geriatric syndromes. * *p* < 0.05. IADL: instrumental activities of daily living and ADL: activities of daily living.

**Figure 2 ijerph-21-00812-f002:**
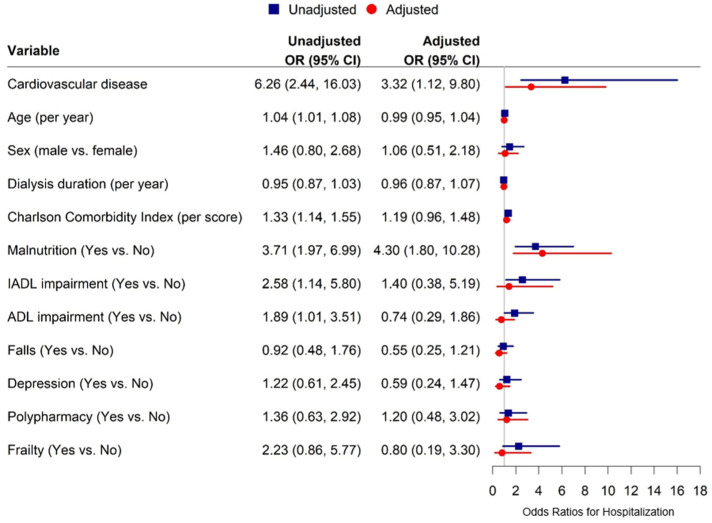
Predictive factors for 6-month hospitalization. IADL: instrumental activities of daily living and ADL: activities of daily living.

**Table 1 ijerph-21-00812-t001:** Participant characteristics.

Variables		N = 175
Age, years		72.4 ± 8.5
Sex		
	Male	72 (41.1%)
	Female	103 (58.9%)
Education		
	Illiterate	6 (3.4%)
	Primary school	48 (27.4%)
	Secondary school	43 (24.6%)
	High school	41 (23.4%)
	Higher education	37 (21.1%)
Living status		
	Alone	6 (3.4%)
	With family	169 (96.6%)
Smoking		50 (28.6%)
Body mass index:		
	Underweight (<18.5)	40 (22.9%)
	Normal (18.5–22.9)	84 (48.0%)
	Overweight (23.0–24.9)	18 (10.3%)
	Obese (≥25.0)	33 (18.9%)
Duration of dialysis, years		3.6 ± 3.5
Charlson Comorbidity Index		8.0 ± 2.3
ADL score		4.4 ± 2.3
IADL score		3.4 ± 3.1
MNA-SF score		8.7 ± 3.6
CFS score		5.4 ± 1.4
GDS-15 score		5.6 ± 3.9
Total number of medications		7.1 ± 1.9
Having ≥1 cardiovascular disease		140 (80.0%)
	Heart failure	105 (60.0%)
	Coronary heart disease	87 (49.7%)
	Stroke	45 (25.7%)
Other chronic health conditions:		
	Anemia	144 (82.8%)
	Diabetes	110 (62.9%)
	Dyslipidemia	103 (58.9%)
	Chronic stomach problems (reflux, heartburn, or gastric ulcer)	42 (24.0%)
	Chronic liver disease	25 (14.3%)
	Chronic obstructive pulmonary disease	14 (8.0%)
	Cancer	7 (4.0%)
Serum laboratory parameters:		
	White blood cells (×10^9^/L)	7.1 ± 2.4
	Hemoglobin (g/dL)	10.5 ± 1.8
	Hematocrit (%)	32.3 ± 5.4
	Platelets (×10^9^/L)	206.6 ± 71.9
	Creatinine (µmol/L)	511.4 ± 219.3
	AST (units/L)	22.5 ± 17.5
	ALT (units/L)	15.3 ± 14.3
	Sodium (mmol/L)	134.5 ± 3.2
	Potassium (mmol/L)	3.5 ± 0.5
	Chloride (mmol/L)	98.3 ± 3.5
	Total calcium (mmol/L)	2.3 ± 0.3
	Albumin (g/L)	35.3 ± 5.1

Continuous data are presented as mean (standard deviation) and categorical data are shown as N (%). ADL: activities of daily living. CFS: Clinical Frailty Scale. IADL: instrumental activities of daily living; GDS: Geriatric Depression Scale. MNA-SF: Mini Nutritional Assessment Short Form.

**Table 2 ijerph-21-00812-t002:** Geriatric syndromes in participants with and without cardiovascular disease.

Geriatric Syndromes	All Participants with End-Stage Renal Disease and Chronic Dialysis (N = 175)	Without Cardiovascular Disease (N = 35)	With Cardiovascular Disease (N = 140)	*p*-Values
Frailty	153 (87.4%)	25 (71.4%)	128 (91.4%)	0.001
Polypharmacy	142 (81.1%)	25 (71.4%)	117 (83.6%)	0.100
IADL impairment	142 (81.1%)	21 (60.0%)	121 (86.4%)	<0.001
Malnutrition	75 (42.9%)	7 (20.0%)	68 (48.6%)	0.004
ADL impairment	65 (37.1%)	6 (17.1%)	59 (42.1%)	0.006
Falls	51 (29.1%)	7 (20.0%)	44 (31.4%)	0.183
Depression	42 (24.0%)	5 (14.3%)	37 (26.4%)	0.132

Categorical data are shown as N (%). ADL: activities of daily living. IADL: instrumental activities of daily living.

## Data Availability

The data presented in this study are available on request from the corresponding author.

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
