# Peer review of "The Burden of Cardiovascular Disease and Geriatric Syndromes in Older Patients Undergoing Chronic Hemodialysis"

_ijerph, 2024, doi:10.3390/ijerph21060812_

Round 1

Reviewer 1 Report

Comments and Suggestions for Authors

This study examines the relationship of cardiovascular disease in patients on chronic kidney hemodialysis and geriatric syndromes.  

The introduction is concise and supports the aims of the study.  References are relevant and timely.

The methods section clearly describes the design, population, variables and measures (used standardized measures), and the statistical plan.

Results are clearly presented both in the tables and body of the manuscript.  I have only one question here: It appears based on Table 1 that a number of participants had multiple CVD diagnoses.  Did the authors consider if having more than one CVD diagnosis could influence the hospitalization rates?  Could you weight the diagnosis in terms of severity (impact on ADL, IADL etc) and examine the impact on hospitalization?  May not be possible at this stage but suggest you consider in the future.

Discussion is relevant.  The non-adherence is very concerning and really needs exploring.  Insightful to say that CVD management is not very good in this particular patient population and really needs more research.

Conclusions are supported.

Author Response

This study examines the relationship of cardiovascular disease in patients on chronic kidney hemodialysis and geriatric syndromes.  

The introduction is concise and supports the aims of the study.  References are relevant and timely.

The methods section clearly describes the design, population, variables and measures (used standardized measures), and the statistical plan.

Responses: Thank you for the time spent reviewing our manuscript and for your comments.

Results are clearly presented both in the tables and body of the manuscript.  I have only one question here: It appears based on Table 1 that a number of participants had multiple CVD diagnoses.  Did the authors consider if having more than one CVD diagnosis could influence the hospitalization rates?  Could you weight the diagnosis in terms of severity (impact on ADL, IADL etc) and examine the impact on hospitalization?  May not be possible at this stage but suggest you consider in the future.

Responses: Thank you for your suggestion. In this study, 45.7% of the participants had 2 or more CVDs. However, this is not the scope of the current analysis. We will consider this for future studies.

Discussion is relevant.  The non-adherence is very concerning and really needs exploring.  Insightful to say that CVD management is not very good in this particular patient population and really needs more research.

Conclusions are supported.

Responses: Thank you for your comments!

Reviewer 2 Report

Comments and Suggestions for Authors

In the manuscript by Nguyen et al., the authors address a pertinent topic concerning cardiovascular disease and geriatric syndromes (frailty, malnutrition, impairment in activities of daily living, impairment in instrumental activities of daily living, depression, falls, and polypharmacy) in 175 elderly patients undergoing chronic hemodialysis, and the associated risk of hospitalization.

The study makes a valuable contribution to the field. Before we proceed, please address the following points:   It would be insightful to provide statistical analyses regarding the presence of comorbidities, such as diabetes mellitus, which increase the risk of micro- and macroangiopathic complications, and their consequent impact on quality of life and frailty.

Likewise, include information on laboratory parameters that may adversely affect the clinical setting, such as the presence of uremia, potassium control, hemodialysis adequacy, bone mineral parameters, among others.

There is room to improve the English grammar and style.

Author Response

Thank you for the time spent reviewing our manuscript and for your comments. We have carefully responded to the issues raised as detailed below.

In the manuscript by Nguyen et al., the authors address a pertinent topic concerning cardiovascular disease and geriatric syndromes (frailty, malnutrition, impairment in activities of daily living, impairment in instrumental activities of daily living, depression, falls, and polypharmacy) in 175 elderly patients undergoing chronic hemodialysis, and the associated risk of hospitalization.

The study makes a valuable contribution to the field. Before we proceed, please address the following points:  

It would be insightful to provide statistical analyses regarding the presence of comorbidities, such as diabetes mellitus, which increase the risk of micro- and macroangiopathic complications, and their consequent impact on quality of life and frailty.

Response: We have now provided more data on other comorbidities in Table 1. Regarding statistical analyses, we used the Charlson Comorbidity Index instead of individual comorbidity. There are 19 conditions in the Charlson Comorbidity Index, including diabetes.

Likewise, include information on laboratory parameters that may adversely affect the clinical setting, such as the presence of uremia, potassium control, hemodialysis adequacy, bone mineral parameters, among others.

Response: We have now provided more data on the laboratory parameters in Table 1.

Reviewer 3 Report

Comments and Suggestions for Authors

The authors present a study that was conducted to understand the prevalence and impact of cardiovascular disease (CVD) and geriatric syndromes in older adults undergoing chronic hemodialysis. This multi-center, observational study included 175 participants aged 65 and above, recruited from two major hospitals in Vietnam between November 2020 and June 2021. The follow-up duration is very short to account for CVD/ and risk of hospitalization, I have some other comments for the authors:

-          Clearly define the age range considered as "older adults" in the study. For instance, specify if it includes patients aged 65 and above.

-          BMI (Obesity Threshold): Include a reference for the threshold of Body Mass Index (BMI) categories used in the study

COVID-19 Admissions/Tests: Mention whether the study accounted for COVID-19 admissions or conducted any COVID-19 tests among participants, as the pandemic might have impacted hospitalization rates and health outcomes.

Type of Vaccinations Provided: Specify if any vaccinations (e.g., influenza, pneumococcal, COVID-19) were considered or recorded as part of the patients' health profiles. Esp. that some vaccines have been involved recently in CVD events.

Alcohol Use: Include data on alcohol consumption among participants, as it could influence both CVD and geriatric syndrome outcomes.

Concurrent Comorbidities: Provide detailed information on other comorbid conditions that were prevalent among the study participants, beyond the Charlson Comorbidity Index.

Blood Pressure Measurements: Mention how blood pressure was monitored and reported, as it is a crucial factor in CVD and overall health assessment, esp. Falls.

Number of Dialysis Sessions/ Vintage of Dialysis: Detail the frequency and duration of dialysis sessions for participants, as these factors can significantly impact their health outcomes.

Vascular Access Type: State the types of vascular access used (e.g., arteriovenous fistula, graft, or catheter), as this can affect dialysis efficacy and complication rates.

Specify the underlying causes of chronic kidney disease (CKD) in the study population, as this might influence both CVD prevalence and geriatric syndromes.

Albumin Levels: Report serum albumin levels, as they are a marker of nutritional status and can be linked to both CVD and geriatric syndromes.

Anemia: Include information on the presence and management of anemia among participants, as it is a common issue in patients with CKD and can influence hospitalization rates.

Description of Tests Used: Provide a brief description of the tests and assessments used to evaluate geriatric conditions, CVD, and other health parameters. This could include specific scales or diagnostic criteria. Consider providing detailed methodologies and assessment tools/tests used in the supplementary materials to ensure transparency and reproducibility.

Author Response

We would like to thank you for your useful comments and suggestions. We have carefully responded to the issues raised as detailed below.

The authors present a study that was conducted to understand the prevalence and impact of cardiovascular disease (CVD) and geriatric syndromes in older adults undergoing chronic hemodialysis. This multi-center, observational study included 175 participants aged 65 and above, recruited from two major hospitals in Vietnam between November 2020 and June 2021. The follow-up duration is very short to account for CVD/ and risk of hospitalization, I have some other comments for the authors:

Clearly define the age range considered as "older adults" in the study. For instance, specify if it includes patients aged 65 and above.

Response: We have added this definition (aged 60 or older) in line 79.

BMI (Obesity Threshold): Include a reference for the threshold of Body Mass Index (BMI) categories used in the study.

Response: We have added the reference:

WHO Expert Consultation. Appropriate body-mass index for Asian populations and its implications for policy and intervention strategies. Lancet. 2004 Jan 10;363(9403):157-63. doi: 10.1016/S0140-6736(03)15268-3

COVID-19 Admissions/Tests: Mention whether the study accounted for COVID-19 admissions or conducted any COVID-19 tests among participants, as the pandemic might have impacted hospitalization rates and health outcomes.

Response: Thank you for your suggestion, but we don’t have these data.

Type of Vaccinations Provided: Specify if any vaccinations (e.g., influenza, pneumococcal, COVID-19) were considered or recorded as part of the patients' health profiles. Esp. that some vaccines have been involved recently in CVD events.

Response: Thank you for your suggestion, but we don’t have these data.

Alcohol Use: Include data on alcohol consumption among participants, as it could influence both CVD and geriatric syndrome outcomes.

Response: Thank you for your suggestion. Unfortunately, we did not collect information about alcohol consumption among the participants.

Concurrent Comorbidities: Provide detailed information on other comorbid conditions that were prevalent among the study participants, beyond the Charlson Comorbidity Index.

Response: We have provided more details of comorbidities in Table 1.

Blood Pressure Measurements: Mention how blood pressure was monitored and reported, as it is a crucial factor in CVD and overall health assessment, esp. Falls.

Response: Thank you for your suggestion. Unfortunately, we did not collect information on blood pressure measures among the participants.

Number of Dialysis Sessions/ Vintage of Dialysis: Detail the frequency and duration of dialysis sessions for participants, as these factors can significantly impact their health outcomes.

Response: Thank you for your suggestion. Unfortunately, we did not collect this information.

Vascular Access Type: State the types of vascular access used (e.g., arteriovenous fistula, graft, or catheter), as this can affect dialysis efficacy and complication rates.

Response: Thank you for your suggestion. Unfortunately, we did not collect this information.

Specify the underlying causes of chronic kidney disease (CKD) in the study population, as this might influence both CVD prevalence and geriatric syndromes.

Response: Thank you for your suggestion. Unfortunately, we did not collect this information.

Albumin Levels: Report serum albumin levels, as they are a marker of nutritional status and can be linked to both CVD and geriatric syndromes.

Response: We have provided albumin levels in Table 1

Anemia: Include information on the presence and management of anemia among participants, as it is a common issue in patients with CKD and can influence hospitalization rates.

Response: We have the prevalence of anemia in Table 1. We did not collect data related to the management of anemia.

Description of Tests Used: Provide a brief description of the tests and assessments used to evaluate geriatric conditions, CVD, and other health parameters. This could include specific scales or diagnostic criteria. Consider providing detailed methodologies and assessment tools/tests used in the supplementary materials to ensure transparency and reproducibility.

Response: We already described them in the manuscript (lines 89-104).

“CVD was defined based on the medical records and as having one of these conditions: heart failure, ischemic heart disease, and stroke.

Comorbidities were assessed using the Charlson Comorbidity Index.15 The diagnosis of CVD and other comorbidities conformed to the International Classification of Diseases, Tenth Revision.

Geriatric syndromes: Participants were assessed for geriatric conditions such as frailty, malnutrition, impairment in activities of daily living (ADL), impairment in instrumental activities of daily living (IADL), depression, falls, and polypharmacy. Frailty was defined according to the Clinical Frailty Scale (CFS), and participants with a CFS total score ≥4 were classified as being frail.16 Malnutrition was defined as a total score ≤ 7 from the Mini Nutritional Assessment Short Form.17 Impairment in activities of daily living was defined as an ADL total score <6.18 Impairment in instrumental activities of daily living was defined as an IADL total score <8.18,19 History of falls was documented from the medical records. Polypharmacy was defined as the concurrent use of ≥ 5 medications. Depression was defined based on the 15-item Geriatric Depression Scale (GDS) and participants with a GDS total score ≥10 were classified as having depression.20”

Round 2

Reviewer 3 Report

Comments and Suggestions for Authors    

As most of the variables are not available, the authors should include the following limitations in the study. Firstly, data regarding COVID-19 admissions and tests were not available, which may affect the admission rates. Additionally, information on the types of vaccinations administered was not included, potentially contributing to cardiovascular events. The absence of data on alcohol consumption, which could influence both CVD and geriatric syndrome outcomes, should also be noted. Furthermore, blood pressure measurements were not included. The study also lacked information on the number of dialysis sessions, vintage of dialysis, and vascular access, which could affect dialysis efficacy and complication rates. Lastly, details on the underlying causes of chronic kidney disease (CKD) in the study population were not provided, which might influence both CVD prevalence and geriatric syndromes. Acknowledging these limitations will provide a clearer understanding of the study's constraints and highlight areas for future research.

Author Response

We would like to thank you for your suggestion.

We have added these sentences in the Limitation section of the manuscript:

"The unavailability of data regarding COVID-19 admissions and tests may have significant implications for the reported admission rates. Additionally, we did not include details on the vaccinations administered, a factor that might contribute to cardiovascular events but remains unexplored within our dataset. Moreover, the study lacks data on alcohol consumption, which is a crucial variable given its potential influence on CVD and geriatric syndrome outcomes. Data on blood pressure measurements were also not collected, depriving us of critical insights into a key risk factor for cardiovascular and other health complications. In the context of dialysis, several important aspects were not recorded. These include the number of dialysis sessions, the vintage of dialysis, and the type of vascular access used. These factors are instrumental in determining dialysis efficacy and the incidence of related complications. Finally, the underlying causes of chronic kidney disease in our study population were not specified. This omission is particularly significant, as different etiologies of chronic kidney disease can have varying impacts on CVD prevalence and the manifestation of geriatric syndromes. By acknowledging these limitations, we can identify areas for future research to enhance the robustness and applicability of our results."